# INTRINSIC MESH CNNS

## ABSTRACT

Rephrasing the convolution operation from Euclidean to non-Euclidean domains, such as graphs and surfaces, is of great interest in the context of geometric deep learning. By elaborating on closing a theoretical gap between an existing framework for the parametric construction of non-Euclidean convolutions and a sound theoretical definition for intrinsic surface convolutions, motivated by differential geometry, we show that existing definitions for surface convolutions only differ in their prior assumptions about local surface information. In the course of our efforts we found a canonical prior that allows for a theoretical definition of the class of *Intrinsic Mesh CNNs*, which captures the CNNs that operate on surfaces. This class combines the practical advantages of the framework for the parametric construction of non-Euclidean convolutions with a substantiated theory, that allows for further theoretical analysis and interesting research questions. Eventually, we conduct an experimental investigation of the canonical prior, the results of which confirm our theory about its canonical nature.

## 1 INTRODUCTION

It is widely known that convolutional neural networks achieve astonishing performances in problem domains such as computer vision (He et al., 2016; Redmon et al., 2016). However, the traditional definition of the convolution operation is limited to Euclidean domains. The growing interest in geometric deep learning has shown that non-Euclidean data is ubiquitous in daily life (Wu et al., 2020; Cao et al., 2020). Besides the recent efforts to extensively investigate graph neural networks, the problem of learning intrinsic surface properties with surface convolutions has attracted a considerable amount of interest (Masci et al., 2015; Boscaini et al., 2016a; Monti et al., 2017). The surface's non-Euclidean nature, however, requires the traditional definition of convolutions to be revised such that it pays attention to intrinsic surface properties. A lot of work on learning intrinsic surface properties focuses on the shape correspondence problem (Masci et al., 2015; Boscaini et al., 2016a; Monti et al., 2017; Poulenard & Ovsjanikov, 2018), which portrays an underlying task to a variety of higher-level problems from computer graphics such as space-time registration and further semantic shape analysis Van Kaick et al. (2011). From the perspective of the Machine Learning community, it is also worth mentioning that it is thinkable to use intrinsic surface convolutions for representation learning and generative models analogously to traditional convolutions on Euclidean data Kingma & Welling (2013); Goodfellow et al. (2020); Ho et al. (2020).

The first work for intrinsic surface convolutions is the one from Masci et al. (2015), who have introduced *geodesic convolutions on Riemannian manifolds* by employing the so called *patch operator*. However, the algorithmic construction of the patch operator involved the computation of so called *local geodesic polar coordinate systems*, which are limited in their extension on the surface. This is why Boscaini et al. (2016a) proposed *anisotropic convolutions* on surfaces which overcome the limiting radius of the mentioned coordinate systems by rephrasing the patch operator into considering spectral properties of the information on the surface. Monti et al. (2017) proposes a general framework that defines *mixture model networks* which operate in non-Euclidean domains such as graphs and surfaces. For example, geodesic- and anisotropic convolutions are obtained as particular instances of that framework. An exceptionally profound overview of the subject of learning in non-Euclidean domains is given in Bronstein et al. (2021), where a detailed insight into the derivation of intrinsic manifold convolutions is given by formulating it as a particular instance of a geometric deep learning blueprint.

This paper elaborates on three aspects. First, we close the theoretical gap between the algorithmic framework of Monti et al. (2017) and the theory grounded definition of Bronstein et al. (2021) for intrinsic surface convolutions and by that see that previous definitions on intrinsic surface convolutions implicitly made use of what we call *priors*. Second, we see that those priors give rise to a notion of *learnable features* for intrinsic surface convolutions. We use these as a means to characterize priors in order to analyse their comprehensiveness. Third, we see that the prior which is required for the connection of the framework from Monti et al. (2017) with the theory of Bronstein et al. (2021) is a very general one. We then make use of our findings and, to be consistent with the nomenclature of Bronstein et al. (2021), give a theoretical grounded definition of the class of *Intrinsic Mesh CNNs* (IMCNNs). Eventually, we see that the results of an experimental evaluation of different IMCNNs supports the theory of this paper.

## 2 BACKGROUND

### 2.1 GEODESIC CONVOLUTION

The adaption of Euclidean convolutions to convolutions in compact Riemannian manifolds has first been made by Masci et al. (2015). For this, they compute local *geodesic polar coordinate systems* (GPC-systems) on the surface, which consist of radial geodesics (rays) and angular level sets (concentric circles). These coordinates are required for the so called *patch operator*, which represents a function that extracts signal values for a point $x$ from the surface:

$$[D(x)s](\rho, \theta) = \int_X v_{\rho,\theta}(x, x') s(x') dx'$$

Here, the $v_{\rho,\theta}(x, x')$ portray interpolation weights and $s(\cdot)$ the signal on the surface. Masci et al. (2015) choose $v_{\rho,\theta}(x, x')$ to be proportional to a two-dimensional Gaussian, which is defined over the geodesic polar coordinates of the pre-computed GPC-systems. Eventually, the *geodesic convolution* in the point $u$ on the surface is defined as:

$$(s * t)_{\Delta\theta}(x) = \sum_{\rho} \sum_{\theta} t(\rho, \theta + \Delta\theta)[D(x)s](\rho, \theta)$$

The "$\Delta\theta$"-term is added because during the construction of the GPC-systems we need to select a reference direction. That direction can be chosen arbitrarily. This problem is referred to as the *angular coordinate ambiguity*. Masci et al. (2015) compute the geodesic convolution for multiple "$\Delta\theta$" and select the result which yields the largest response. This process is referred to as *angular max-pooling*.

### 2.2 ANISOTROPIC CONVOLUTION

In addition to the angular coordinate ambiguity problem, GPC-systems suffer from being limited by the so called *injectivity radius*. This is why Boscaini et al. (2016a) propose a different way of extracting features from the surface. Therefore, they consider the *anisotropic heat equation*:

$$\frac{\partial}{\partial\tau} s(\tau, x) = -\Delta_{\alpha\theta} s(\tau, x)$$

where $s$ describes the heat at $x$ at time $\tau$ and $\Delta_{\alpha\theta}$ the *anisotropic Laplacian*, which considers a *conductivity* $\alpha$ and a rotation $\theta$ w.r.t. the *maximum curvature* of the surface at $x$. Its exact definition is given in the appendix. This rotation to a so called *fixed gauge* shall resolve the angular coordinate ambiguity. The anisotropic diffusion equation can be solved by "applying" the *anisotropic heat kernel* onto an initial solution $s(0, x)$ for the anisotropic heat equation. Thereby, the anisotropic heat kernel is defined as:

$$h_{\alpha\theta\tau}(x, y) = \sum_{n} e^{-\tau\lambda_{\alpha\theta n}} \phi_{\alpha\theta n}(x)\phi_{\alpha\theta n}(y)$$

where $\{\phi_{\alpha\theta n}\}_n$ are the Eigenfunctions of $-\Delta_{\alpha\theta}$ for the Eigenvalues $\{\lambda_{\alpha\theta n}\}_n$. Boscaini et al. (2016a) use the anisotropic heat kernels to define the patch operator in the spectral domain:

$$[D_\alpha(x)s](\tau, \theta) = \frac{\int_X h_{\alpha\theta\tau}(x, y)s(y)\, dy}{\int_X h_{\alpha\theta\tau}(x, y)\, dy}$$

Eventually, Boscaini et al. (2016a) use this patch operator to define the *anisotropic convolution*:

$$(s * t)(x) = \int k(\tau, \theta)[D_\alpha(x)s](\tau, \theta) \, d\tau d\theta$$

## 2.3 MIXTURE OBJECT NETWORKS

Monti et al. (2017) generalizes the attempts of Masci et al. (2015) and Boscaini et al. (2016a) by proposing a general framework for defining non-Euclidean convolutions in domains such as graphs and manifolds. This framework introduces a parametric construction of the patch operator via so called *pseudo coordinates* $\boldsymbol{u}(x, y)$ and *kernels* $w_j(\boldsymbol{u}(x, y))$. In particular, their general patch operator has the form:

$$[D(x)s](j) = \sum_{y \in \mathcal{N}(x)} w_j(\boldsymbol{u}(x, y))s(y), \ j = 1, ..., J$$

where $x$ portrays a point in the respective domain and $\mathcal{N}(x)$ a neighborhood of $x$. In case of the domain being a continuous manifold, the sum should be interpreted as an integral. The final convolution then uses the parametric patch operator:

$$(s * t)(x) = \sum_{j=1}^{J} t(j)[D(x)s](j)$$

Thereby, the framework does not only allow for the parametric construction of the geodesic- (Masci et al., 2015), or anisotropic convolutional neural networks (Boscaini et al., 2016a), but also for the construction of traditional CNNs (LeCun et al., 1998) in the Euclidean domain, graph convolutional neural networks (Kipf & Welling, 2016) or diffusion convolutional neural networks (Atwood & Towsley, 2016).

## 2.4 CONVOLUTIONS ON A MANIFOLD

A less algorithmic and a more theory grounded perspective on intrinsic surface convolutions is given by Bronstein et al. (2021). They motivate intrinsic surface convolutions with the help of differential geometry. Traditionally, convolutions between a signal $s$ and a template $t$ in a point $\boldsymbol{u}$ are defined in a Euclidean domain:

$$(s * t)(\boldsymbol{u}) = \int_{\mathbb{R}^n} s(\boldsymbol{v})t(\boldsymbol{u} - \boldsymbol{v})d\boldsymbol{v}$$

The convolution shifts the template $t$ into point $\boldsymbol{u}$ and accesses the weights of $t$ relative to the point $\boldsymbol{u}$ by computing $\boldsymbol{u} - \boldsymbol{v}$. Thereby, $\boldsymbol{u} - \boldsymbol{v}$ yields a vector that points from $\boldsymbol{v}$ to $\boldsymbol{u}$. This vector exhibits a notion of relative direction between $\boldsymbol{u}$ and $\boldsymbol{v}$. In general compact Riemannian manifolds $M^n$, however, subtraction is undefined. Instead, if we want to compute the convolution in point $\boldsymbol{u} \in M^n$, we make use of *tangent vectors* $\boldsymbol{y} \in T_{\boldsymbol{u}}M^n$ from the tangent space $T_{\boldsymbol{u}}M^n$ at $\boldsymbol{u}$, which locally exhibit a notion of direction. Due to the tangent vectors $\boldsymbol{y}$ being coordinate free in general, we need to choose a basis for the tangent space in order to be able to calculate with $\boldsymbol{y}$. This basis is given by a frame called *gauge* $\omega_{\boldsymbol{u}}$, that can be considered a map which defines a basis for each tangent space $T_{\boldsymbol{u}}M^n$. Yet, multiple gauges are possible for one tangent space. Different $\omega_{\boldsymbol{u}}$ cause different coordinates, which in turn cause different results in the convolution. This represents the theoretical link to the aforementioned angular coordinate ambiguity problem. Sophisticated solutions to this problem lead to the topic of *gauge-equivariant convolutions* on compact Riemannian manifolds (Bronstein et al., 2021; Cohen et al., 2019; De Haan et al., 2020). However, a detailed review of those would exceed the boundaries of this work.

While the tangent vectors $\boldsymbol{y}$ yield a helpful means to describe a local notion of direction, they do not represent the elements of the surface on which the signal $s$ is defined. The *exponential map* $\exp_u : T_u M^n \to M^n$ portrays a local diffeomorphism, limited by the previously discussed injectivity radius, that maps tangent vectors onto elements of the manifold.

Eventually, Bronstein et al. (2021) connects the gauge $\omega_u$, which allows us to use coordinates to reference certain tangent vectors, the exponential map, which associates the directions locally with points on the manifold, and the signal of interest, which is defined on the manifold, to one function in order to define the intrinsic convolution in manifolds:

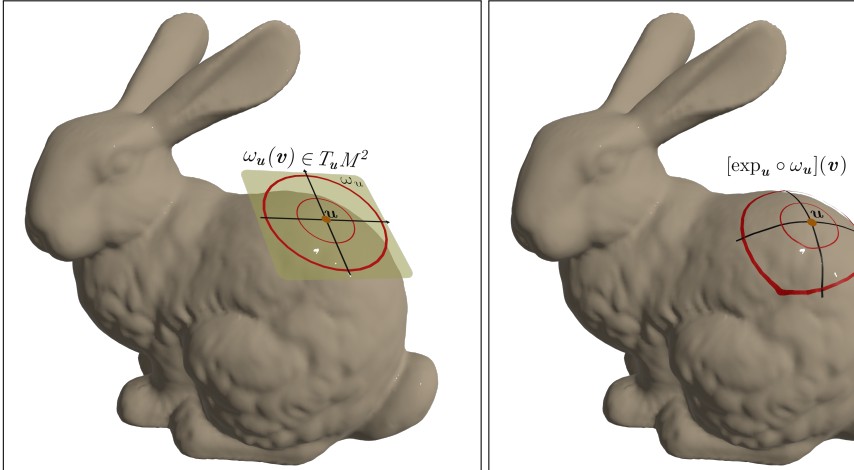

Figure 1: Exemplary illustration of $[s \circ \exp_u \circ \omega_u]$ on the Stanford Bunny. **[Left]** In order to describe relative positions around $\boldsymbol{u} \in M^2$ we consider the tangent vectors $\boldsymbol{y}$ in tangent plane $T_{\boldsymbol{u}}M^2$. We choose a basis in form of a coordinate frame via the gauge $\omega_{\boldsymbol{u}}$ within the tangent plane $T_{\boldsymbol{u}}M^2$ to access the tangent vectors. There is no unique gauge. That is, other gauges, e.g. $\omega_u$ that give rise to frames with a different orientation within $T_{\boldsymbol{u}}M^2$, are valid choices. **[Right]** We locally map the tangent vectors $\omega_{\boldsymbol{u}}(\boldsymbol{v}) = \boldsymbol{y} \in T_{\boldsymbol{u}}M^2$ at coordinates $\boldsymbol{v} \in [0,1]^2$ into the surface with the exponential map $\exp_{\boldsymbol{u}}$. The signal, e.g. local surface descriptors such as SHOT Tombari et al. (2010) or Optimal Spectral Descriptors Litman & Bronstein (2013), is defined on the surface. Thus, given $\exp_{\boldsymbol{u}}(\omega_{\boldsymbol{u}}(\boldsymbol{v})) = \boldsymbol{w} \in M^2$, we can now extract the surface signal by calculating $s(\boldsymbol{w})$.

**Definition 1** (Intrinsic Manifold Convolution (Bronstein et al., 2021)). *The intrinsic manifold convolution of a signal $s : M^n \to \mathbb{R}$ defined on the $n$-dimensional compact Riemannian manifold $M^n$ with a template $t : \mathbb{R}^n \to \mathbb{R}$ in point $\boldsymbol{u} \in M^n$ is defined as:*

$$(s * t)(\boldsymbol{u}) = \int\limits_{[0,1]^n} t(\boldsymbol{v}) \, [s \circ exp_u \circ \omega_u](\boldsymbol{v}) \, d\boldsymbol{v}$$

In the case of computing convolutions on a 2-dimensional, compact Riemannian manifold $M^2$ we refer to it as the *intrinsic surface convolution* (ISC). In that case, the unit-cube $[0, 1]^2$ is homeomorphic to the affine tangent plane attached to the point $\boldsymbol{u} \in M^2$. This allows us to visually think of extracting local features of the manifold into the tangent plane $T_{\boldsymbol{u}}M^2$ and conducting the convolution within said tangent plane. See Figure 1 for a visualization.

## 3    INTRODUCING DIRAC TO INTRINSIC SURFACE CONVOLUTIONS

In the previous section we have discussed algorithmic (Masci et al., 2015; Boscaini et al., 2016a; Monti et al., 2017) and mathematical (Bronstein et al., 2021) approaches to intrinsic surface convolutions. In this section we bridge the theoretical gap between the framework of Monti et al. (2017) and the theoretical definition for intrinsic surface convolutions from Bronstein et al. (2021) by first reformulating the non-Euclidean convolution equation of Monti et al. (2017) into the definition of Bronstein et al. (2021) and subsequently introducing a previously unused kernel to the framework. Due to the reformulation we witness two major insights. First, the introduction of the patch operator by Masci et al. (2015) implicitly gives rise to a notion of *learnable features* and they dependent on a selected *prior*. Second, the mathematically motivated intrinsic surface convolution by Bronstein et al. (2021) only differs in its kernel to the geodesic- (Masci et al., 2015) and anisotropic convolution (Boscaini et al., 2016a). We begin this section by unifying the previous definitions for intrinsic surface convolutions.

**Theorem 1.** *Let $p \in C^0(\mathbb{R}^n \times \mathbb{R}^n)$ be a kernel in the sense of Monti et al. (2017), $B_R(\boldsymbol{0}) \subset \mathbb{R}^2$ the disc with radius $R$ around $\boldsymbol{0}$ and*

$$[D_p(\boldsymbol{u})s](\boldsymbol{v}) = \int\limits_{B_R(\boldsymbol{0})} p_{\boldsymbol{v}}(\boldsymbol{y})[s \circ exp_{\boldsymbol{u}} \circ \omega_{\boldsymbol{u}}](\boldsymbol{y}) \, d\boldsymbol{y}$$

*the continuous version of the parametric patch operator from Monti et al. (2017). For a continuous function $t \in C^0(\mathbb{R}^n)$, called the template, we have that:*

$$(s * t)_{\Delta\theta,p}(\boldsymbol{u}) = \int\limits_{B_R(\boldsymbol{0})} t(\boldsymbol{v})[D_p(\boldsymbol{u})s](\boldsymbol{v}) \, d\boldsymbol{v}$$

$$= \int\limits_{B_R(\boldsymbol{0})} \widetilde{p}_t(\boldsymbol{y})[s \circ exp_{\boldsymbol{u}} \circ \omega_{\boldsymbol{u}}](\boldsymbol{y}) \, d\boldsymbol{y} = (s * \widetilde{p}_t)_{\Delta\theta}(\boldsymbol{u})$$

*with $\widetilde{p}_t(\boldsymbol{y})$ being defined as:*

$$\widetilde{p}_t(\boldsymbol{y}) = \int\limits_{B_R(\boldsymbol{0})} t(\boldsymbol{v})p_{\boldsymbol{v}}(\boldsymbol{y}) \, d\boldsymbol{v}$$

We put the proof into the appendix. As we will see in the next section, the choice of $p$ poses a limitation on the features $\widetilde{p}_t(\boldsymbol{y})$ that can be learned by the network. It thus can be used to encode prior knowledge and we therefore refer to it as *prior* and to $\widetilde{p}_t(\boldsymbol{y})$ as *learnable features*.

Using Theorem 1, we can derive the definition for intrinsic surface convolutions of Bronstein et al. (2021) by introducing a previously unused prior for the framework of Monti et al. (2017). Our goal is to specify a prior that yields $\widetilde{p}_t(\boldsymbol{y}) = t(\boldsymbol{y})$. Considering the integral of a continuous function and a normal distribution, we observe that for a diminishing variance, the value of that integral tends towards the value of the function at the mean of the normal. To connect this to the previous theory, we consider the density of that normal distribution as prior $p$. That is, we can achieve our goal by integrating with a normal distribution centered at our interest point $\boldsymbol{y}$:

$$\varphi_{\boldsymbol{x}}^{(n)}(\boldsymbol{y}) = \frac{1}{n\sqrt{2\pi}} e^{-\frac{1}{2}\left(\frac{\|\boldsymbol{x}-\boldsymbol{y}\|}{n}\right)^2}$$

We now formulate our aforementioned intuition about decreasing variances over the limit of the learnable features when using a normal distribution $\varphi_{\boldsymbol{x}}^{(n)}(\boldsymbol{y})$ as a prior:

$$\lim_{n \to 0} \widetilde{\varphi}_t^{(n)}(\boldsymbol{y}) = \lim_{n \to 0} \int\limits_{B_R(\boldsymbol{0})} t(\boldsymbol{v})\varphi_{\boldsymbol{v}}^{(n)}(\boldsymbol{y}) \, d\boldsymbol{v} = t(\boldsymbol{y})$$

This assumes that the point of interest is in the integration domain, i.e., $\boldsymbol{y} \in B_R(\boldsymbol{0})$. In order to get back to our prior notion, we could consider the limit of the normal distributions first, which convergences weakly against the *Dirac distribution* at $\boldsymbol{y}$. By abuse of notation, we will denote this as:

$$\widetilde{\delta}_t(\boldsymbol{y}) = \int\limits_{B_R(\boldsymbol{0})} t(\boldsymbol{v})\delta(\boldsymbol{y} - \boldsymbol{v}) \, d\boldsymbol{v} = t(\boldsymbol{y})$$

and define $\delta(\cdot)$ to be the *Dirac prior*. By inserting the Dirac prior into Theorem 1 we get:

$$(s * \widetilde{\delta}_t)_{\Delta\theta}(\boldsymbol{u}) = \int\limits_{B_R(\boldsymbol{0})} \widetilde{\delta}_t(\boldsymbol{y})[s \circ \exp_{\boldsymbol{u}} \circ \omega_{\boldsymbol{u}}](\boldsymbol{y}) \, d\boldsymbol{y}$$

$$= \int\limits_{B_R(\boldsymbol{0})} \left[ \int\limits_{B_R(\boldsymbol{0})} t(\boldsymbol{v})\delta(\boldsymbol{y} - \boldsymbol{v}) \, d\boldsymbol{v} \right] [s \circ \exp_{\boldsymbol{u}} \circ \omega_{\boldsymbol{u}}](\boldsymbol{y}) \, d\boldsymbol{y}$$

$$= \int\limits_{B_R(\boldsymbol{0})} t(\boldsymbol{y})[s \circ \exp_{\boldsymbol{u}} \circ \omega_{\boldsymbol{u}}](\boldsymbol{y}) \, d\boldsymbol{y}$$

Thus, the definition for intrinsic surface convolutions by Bronstein et al. (2021) can be obtained from the framework of Monti et al. (2017), by using the Dirac prior in the aforementioned sense. This means, that in difference to the previously studied intrinsic surface convolutions like the geodesic-(Masci et al., 2015) or anisotropic convolution (Boscaini et al., 2016a), we now use a different prior. It should be pointed out that in Theorem 1 we have assumed that $p$ has to be continuous. Thus, strictly speaking, we are formally not allowed to simply insert the Dirac distribution into Theorem 1, since the Dirac distribution is no continuous function. While a thorough examination of the relaxation of the continuity assumption would give rise to a larger set of possible priors and by that raises an interesting research question, it exceeds the scope of this work. This is why we leave it open for future work. Nevertheless, despite using a formal approximation to define the Dirac prior, we still can explain why it is interesting. Developing the formalities and understanding why this is the case is the topic of the next section.

## 4 The Class of Intrinsic Mesh CNNs

In the previous section we have closed the theoretical gap between the general framework for non-Euclidean convolutions of Monti et al. (2017) to the theoretically grounded definition for intrinsic surface convolutions by Bronstein et al. (2021) by reformulating the parametric patch operator (Monti et al., 2017) and introducing the Dirac prior. Since priors exhibit a central notion for intrinsic surface convolutions, we dedicate our attention in this section onto the formal characterization of them. Due to our characterization we see that different priors pose different limitations on learnable features. Thereby, the Dirac prior, while being of comparably simple nature, allows to learn very general features making it a suitable canonical choice that allows for a general definition of the class of Intrinsic Mesh CNNs (IMCNNs).

Priors are the only formal difference for different intrinsic surface convolutions. Therefore it is evident that in order to analyse differences between different intrinsic surface convolutions, we should study the differences between their selected priors. To that end, we characterize a prior $p$ by its set of learnable features:

$$\mathbb{F}(p) = \left\{ \widetilde{p}_t(\cdot) \mid t \in C^0(\mathbb{R}^n) \right\} = \left\{ \int_{B_R(\mathbf{0})} t(\mathbf{v}) p_{\mathbf{v}}(\cdot) \, d\mathbf{v} \;\middle|\; t \in C^0(\mathbb{R}^n) \right\}$$

Although this is a very simple characteristic of $p$, it already allows us to tell which priors give rise to more comprehensive intrinsic surface convolutions than others. $\mathbb{F}(\cdot)$ can be used to compare two priors $a$ and $b$ against each other by comparing their sets of learnable features $\mathbb{F}(a)$ and $\mathbb{F}(b)$. For example, if $\mathbb{F}(a) \subsetneq \mathbb{F}(b)$ than we know that we can learn more features with prior $b$ than with prior $a$. In other words, prior $b$ is more comprehensive than prior $a$, if for any learned weights $t_1$ there exist learnable weights $t_2$ such that the resulting learned features are equal, i.e. $\widetilde{a}_{t_1} = \widetilde{b}_{t_2}$:

$$\forall t_1 \in C^0(\mathbb{R}^n) \, \exists t_2 \in C^0(\mathbb{R}^n) \, \forall \mathbf{y} \in \mathbb{R}^n : \int_{B_R(\mathbf{0})} t_1(\mathbf{v}) a_{\mathbf{v}}(\mathbf{y}) \, d\mathbf{v} = \int_{B_R(\mathbf{0})} t_2(\mathbf{v}) b_{\mathbf{v}}(\mathbf{y}) \, d\mathbf{v} \quad (1)$$

The fact, that we can compare priors by comparing their sets of learnable features leads to the following insight:

**Corollary 1.** *Let the set of all priors be given by* $\mathbb{W} = C^0(\mathbb{R}^n \times \mathbb{R}^n)$. *$\mathbb{W}$ has a partial order which is imposed by the subset relation* $\subseteq$ *in the sense that:*

$$a, b \in \mathbb{W} : a \preccurlyeq b :\Leftrightarrow \mathbb{F}(a) \subseteq \mathbb{F}(b)$$

Corollary 1 represents the formalization of our previous intuition, that different priors impose different limitations on the learnable features and therefore can differ in their comprehensiveness. One particularly interesting example is given by our previously introduced Dirac prior. It exhibits a very canonical nature, which is visible by the following two aspects. On the one hand, if we compare it to other priors $a$ via equation 1:

$$\forall t_1 \in L^2(\mathbb{R}^n) \, \exists t_2 \in L^2(\mathbb{R}^n) \, \forall \mathbf{y} \in \mathbb{R}^n : \widetilde{a}_{t_1} = \int_{B_R(\mathbf{0})} t_1(\mathbf{v}) a_{\mathbf{v}}(\mathbf{y}) \, d\mathbf{v} = t_2(\mathbf{y})$$

we see that the Dirac prior allows to learn the features of prior $a$, i.e. $\widetilde{a}_{t_1}$, directly with $t_2$, instead of taking a detour over learning weights $t_1$ to use them in combination with prior $a$ in order to compute suitable features for the convolution. On the other hand, its set of learnable features

$$\mathbb{F}(\delta) = \left\{ \widetilde{\delta}_t(\cdot) \mid t \in C^0(\mathbb{R}^n) \right\} = \left\{ t(\cdot) \mid t \in C^0(\mathbb{R}^n) \right\} = C^0(\mathbb{R}^n)$$

is not limited by an integral and therefore allows to learn comparably many features in contrast to other priors $p$. Due to the Dirac prior's canonical nature we think that it yields a suitable common ground for further research in the realm of intrinsic surface convolutions. This is why we use it to define the class of *Intrinsic Mesh CNNs*:

**Definition 2** (Intrinsic Mesh CNNs). *The class of Intrinsic Mesh CNNs (IMCNNs) is given by the set of convolutional neural networks defined by the intrinsic surface convolutions:*

$$(s * \widetilde{p}_t)_{\Delta\theta}(\boldsymbol{u}) = \int\limits_{B_R(\boldsymbol{0})} \widetilde{p}_t(\boldsymbol{y})[s \circ exp_{\boldsymbol{u}} \circ \omega_{\boldsymbol{u}}](\boldsymbol{y})\, d\boldsymbol{y}$$

*with learned features*

$$\widetilde{p}_t(\boldsymbol{y}) = \int\limits_{B_R(\boldsymbol{0})} t(\boldsymbol{v}) p_{\boldsymbol{v}}(\boldsymbol{y})\, d\boldsymbol{v}$$

*that use priors which admit to learn features that are also learnable with the Dirac prior:*

$$IMCNNs := \left\{ (s * \widetilde{p}_t)_{\Delta\theta}(\boldsymbol{u}) \mid p \preccurlyeq \delta \right\}$$

In the next section of this work, we conduct a variety of experiments to empirically study the performance of different IMCNNs. Thereby, we lie our focus on the comparison of the IMCNN that uses the Dirac prior by comparing it to IMCNNs which use other priors.

## 5 EXPERIMENTAL EVALUATION OF PRIORS

In the last section we have formally investigated priors by characterizing them with their sets of learnable features. Furthermore, we have seen that the set of all priors has a partial order which is imposed by the subset relation given by the different sets of learnable features, meaning that different priors pose different limitations on what features the network can learn. Lastly, we gave a definition for the class of Intrinsic Mesh CNNs with the help of the canonical nature of the Dirac prior. In this section we practically investigate our theory by conducting several experiments with different IMCNNs. By witnessing different performances for different IMCNNs we see that the experiments support our theory, that different priors pose different limitations for what an IMCNN can learn.

In our experiments, we will compare the performance of IMCNNs for the (full) shape correspondence problem. The shape correspondence problem is thoroughly discussed in the computer vision community (Van Kaick et al., 2011) and can be understood as a multi-class classification problem. The goal is to label a point $\boldsymbol{x}$ from a query shape $\mathcal{Q}$ with index $k$ of the corresponding point $\boldsymbol{y}_k$ on a reference shape $\mathcal{R}$. If $s : \mathcal{Q} \to \mathbb{R}$ is the signal defined on the query shapes $\mathcal{Q}$ of our dataset and assuming $\mathcal{R}$ has $|\mathcal{R}|$ vertices, our IMCNNs shall predict a probability distribution $h(s(\boldsymbol{x})) \in \mathbb{R}^{|\mathcal{R}|}$, sometimes referred to as a soft correspondence (Masci et al., 2015), over all $|\mathcal{R}|$ vertices of the reference shape $\mathcal{R}$. A visual example is provided in Figure 3 in the appendix. Since we have a multi-class classification problem, we are using the categorical cross-entropy as the loss function for our training. Our network architecture considers three intrinsic surface convolutions with intermediate angular max-pooling layers (ISC128+ReLU, AMP, ISC128+ReLU, AMP, ISC128+ReLU, AMP, LIN6890). Each convolution computes 128-dimensional embeddings for all points in the query shape. Besides the Dirac prior, we are also considering the Gaussian prior of the geodesic convolution (Masci et al., 2015), an exponential prior, a $\mathcal{X}^2$-prior and a student-t prior in our experiments. Their definitions are given in the appendix.

For our experiments we use the FAUST dataset (Bogo et al., 2014). The dataset consists of 100 triangle meshes which portray ten human subjects in ten different poses, each one containing 6890 vertices. We split the dataset in accordance to Masci et al. (2015) into a train-, validation and test set. The triangle meshes $0 - 79$ are put into the training set, meshes $70 - 79$ are used for

Table 1: Configuration of the used hyperparameters for the conducted experiments.

| Template Discretization | $\rho_0 \approx 0.028$ | $N_\rho = 5$ | $N_\theta = 8$ |
|---|---|---|---|
| GPC-systems | $R \approx 0.037$ | | |
| Optimizer (Adam) | $\gamma \approx 0.0009$ | $\beta_1 = 0.9$ | $\beta_2 = 0.999$ |

validation and $80 - 99$ for testing purposes. Each mesh is shifted such that its centroid is located at $\mathbf{0}$. Subsequently, we uniformly scale each mesh by dividing the vertex coordinates of all dimensions through the geodesic diameter of the mesh to get a maximal geodesic diameter of 1 for all meshes in the dataset.

In order to compute intrinsic surface convolutions, we have to discretize template $t(\cdot)$ and the patch operator $[D(x)s](\rho, \theta)$. Our template discretization is akin to the one proposed in Masci et al. (2015). That is, we discretize $t$ into having $N_\rho$ equi-distant radial level sets with radii $\rho_i = (i + 1)\rho_0/N_\rho$ for $i \geq 0$, with $\rho_0$ being the maximal radial distance, and $N_\theta$ equi-distant angular coordinate rays with angles $\theta_j = 2j\pi/N_\theta$. The Cartesian-product $\mathbb{T} = \{\rho_i\}_{i=0}^{N_\rho-1} \times \{\theta_j\}_{j=0}^{N_\theta-1}$ yields *template vertices*. We now define a tensor $\mathbf{T}_{raxy}$ that associates a trainable weight matrix $\mathbf{T}_{ra} \in \mathbb{R}^{m \times n}$ with each template vertex $(\rho_r, \theta_a) \in \mathbb{T}$. Both, the coordinates $\mathbb{T}$ together with their associated weights $\mathbf{T}$ represent the discretization of $t(\cdot)$.

Next, we discretize the patch operator $[D(x)s](\rho, \theta)$ as follows: First, we compute a GPC-system at each vertex $\boldsymbol{v}_k$ of a mesh $\mathcal{Q}$ with a maximum geodesic radius of $R$, by using the algorithm of Melvær & Reimers (2012). Then we place the template vertices $\mathbb{T}$ into the computed GPC-systems, causing each template vertex to lie in a triangle. Similarly to Poulenard & Ovsjanikov (2018), we now compute the barycentric coordinates of each template vertex in each GPC-system and store these in a tensor $\mathbf{B}_{kraic}$ with $k = 0, ..., |\mathcal{Q}| - 1; r = 0, ..., N_\rho - 1; a = 0, ..., N_\theta - 1; i \in \{0, 1, 2\}$ and $c \in \{0, 1\}$. Thereby, $\mathbf{B}_{krai1}$ contains the $i$-th barycentric coordinate for template vertex $(\rho_r, \theta_a)$ in the GPC-system that has its origin in the $k$-th vertex of $\mathcal{Q}$. $\mathbf{B}_{krai0}$ contains the index of the vertex for the associated barycentric coordinate. In theory, the signal $s : \mathcal{Q} \to \mathbb{R}$ is defined as a scalar function at each point on the surface. In practice, we generalize $s$ to be vector valued, i.e. $s : \mathcal{Q} \to \mathbb{R}^n$. Hence, $s$ is given by a matrix $\boldsymbol{S} \in \mathbb{R}^{|\mathcal{Q}| \times n}$, where the $i$-th row $\boldsymbol{S}_i \in \mathbb{R}^n$ contains the signal for the $\boldsymbol{v}_i$. Lastly, we define a tensor $\mathbf{W}_{raxy}$ that defines the values $p_{(\rho_r, \theta_a)}(\rho_x, \theta_y)$ of our prior. Combining everything to the discretized patch operator yields:

$$[D_{\mathbf{W}}(\boldsymbol{v}_k)\boldsymbol{S}](\rho_r, \theta_a) = \sum_{x=0}^{N_\rho-1} \sum_{y=0}^{N_\theta-1} \mathbf{W}_{raxy} \sum_{i=0}^{2} \mathbf{B}_{kxyi1} \boldsymbol{S}_{\mathbf{B}_{kxyi0}} \in \mathbb{R}^n$$

Figure 4 in the appendix visually helps to get an overview of that process. Given the discretized patch operator, we can now formulate the discretized intrinsic surface convolution as:

$$(\boldsymbol{S} * \mathbf{T})_{\mathbf{W}}(\boldsymbol{v}_k) = \sum_{r=0}^{N_\rho-1} \sum_{a=0}^{N_\theta-1} \mathbf{T}_{ra}[D_{\mathbf{W}}(\boldsymbol{v}_k)\boldsymbol{S}](\rho_r, \theta_a) \in \mathbb{R}^m$$

Similar to Monti et al. (2017), we use 544-dimensional SHOT-descriptors (Tombari et al., 2010) to represent the initial surface signal $\boldsymbol{S}$. In all experiments, we use Adam (Kingma & Ba, 2014) with an equal learning rate $\gamma$, and first and second momentum $\beta_1$ and $\beta_2$ over all experiments. All of the chosen values for our hyperparameters are given in Table 1. We have conducted the experiments using *our library*[1] which implements the neural network layers, all necessary preprocessing procedures and allows the user to easily define and test new priors. In contrast to previous work, we do not post-process the networks results with functional maps (Masci et al., 2015; Boscaini et al., 2016a), intrinsic Bayesian filters Monti et al. (2017) nor any other method.

Figure 2 shows that throughout all conducted experiments the IMCNN that uses the Dirac prior achieves comparable or even better accuracy. On the one hand, this is visible by comparing the exact accuracy, i.e. the point correspondence predictions which are correct and thus yield a geodesic error of zero. With nearly $40\%$ accuracy the IMCNN with the Dirac prior is better than any other observed IMCNN. On the other hand, the graph of the IMCNN with the Dirac prior is typically the

---

[1]*The code can be found in the supplement and will be made public after publication.*

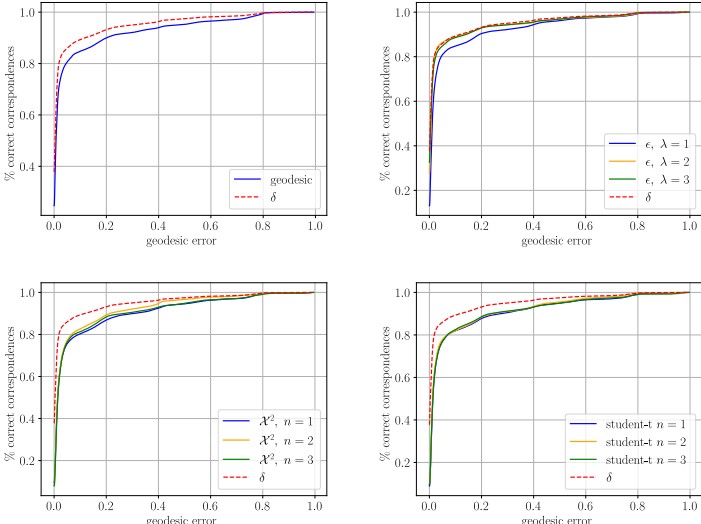

Figure 2: Comparison among training results via the Princeton benchmark (Kim et al., 2011) on the test split of the FAUST dataset. The benchmark captures the accuracy of an IMCNN, which has learned to predict point correspondences. It does so by measuring the geodesic distance or error, respectively, of the predicted vertex to the ground truth vertex. In the plots, the red and dashed graph always represents the accuracy of the IMCNN that uses no prior. The other graphs represent the accuracies of the IMCNNs with priors configured according to the attached legends.

steepest. That means that the incorrect correspondence predictions of the IMCNN with the Dirac prior typically lie closer to the ground truth vertices compared the mispredictions of the IMCNNs with other priors.

That is, our experiments suggest that the IMCNN with the Dirac prior learns features which eventually cause better predictions. We conjecture that we get these results because $\mathbb{F}(\delta)$ is not limited by an integral compared to the $\mathbb{F}(p)$ of the other priors. We thus deem the IMCNN with the Dirac prior to be less error prone than IMCNNs that use a different prior. This is a beneficial insight since it gives rise to the rule of thumb, that we do not have to elaborate on which priors are adequate for a problem and which are not. The IMCNN will probably learn "a more suitable prior" implicitly anyway.

## 6  CONCLUSION

Due to the efforts of this work we can conclude that rephrasing the parametric construction of Monti et al. (2017) into the definition for intrinsic surface convolutions by Bronstein et al. (2021) with the help of the Dirac prior gives rise to the formal class of Intrinsic Mesh CNNs. Intrinsic Mesh CNNs can differ in their comprehensiveness as their assumed priors give rise to different sets of learnable features. The results of our experimental evaluation support the derived theory.

## 7  ACKNOWLEDGEMENTS

*Anonymized due to reviewing purposes.*

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

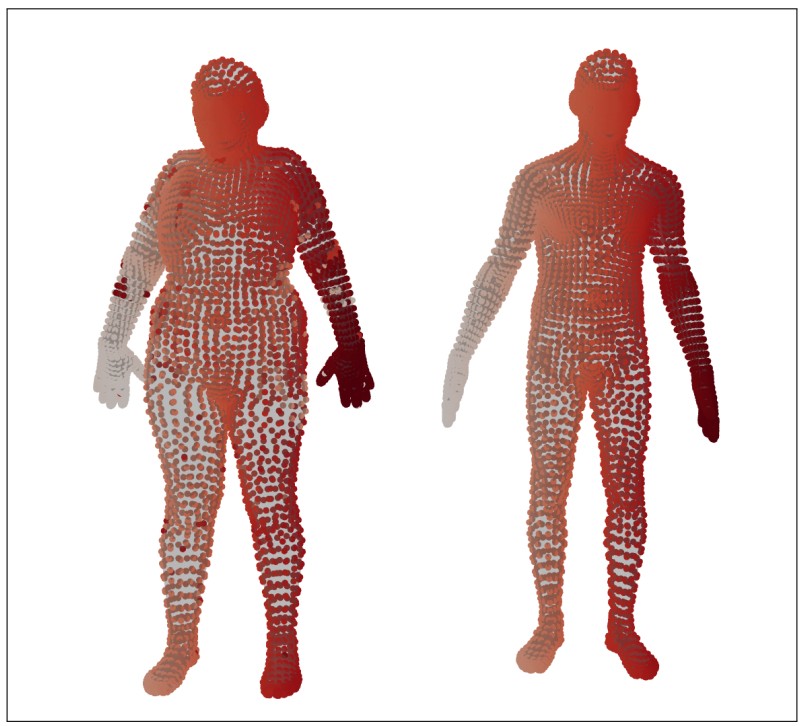

Figure 3: Illustration of the predicted shape correspondence by an IMCNN without post-processing the results, between two normalized meshes from the FAUST-dataset Bogo et al. (2014). The right mesh represents the reference mesh $\mathcal{R}$. The left mesh portrays the query mesh $\mathcal{Q}$. The IMCNN gets a signal $s(\boldsymbol{v})$ and barycentric coordinates defined for the query mesh vertices $\boldsymbol{v} \in \mathcal{Q}$ as input and tries to predict index $k$ of the corresponding vertex $\boldsymbol{y}_k \in \mathcal{R}$ from the reference mesh. The colors of the vertices from the query mesh carry the color of the predicted vertex from the reference mesh. Hence, the color map visualizes the predicted correspondence.

## A    ANISOTROPIC LAPLACIAN

Let $X$ be a Riemannian manifold. Boscaini et al. (2016a) define the anisotropic Laplacian $\Delta_{\alpha\theta}$ as:

$$\Delta_{\alpha\theta} f(x) = -\mathrm{div}_X \left( \boldsymbol{D}_{\alpha\theta}(x) \, \nabla_X f(x) \right)$$

where $\boldsymbol{D}_{\alpha\theta}(x)$ represents the so called *thermal conductivity tensor*:

$$\boldsymbol{D}_{\alpha\theta}(x) = \boldsymbol{R}_\theta(x) \begin{bmatrix} \alpha & \\ & 1 \end{bmatrix} \boldsymbol{R}_\theta^T(x)$$

that uses $2 \times 2$ rotation matrices $\boldsymbol{R}_\theta(x)$. Furthermore, $\nabla_X$ represents the intrinsic gradient (Boscaini et al., 2016a;b):

$$\nabla_X f(x) = \nabla (f \circ \exp_x)(\boldsymbol{0})$$

The function $\mathrm{div}_X$ yields the intrinsic divergence (Boscaini et al., 2016a;b) of a function $f : X \to \mathbb{R}$ and is defined as (Boscaini et al., 2016a;b):

$$\mathrm{div}_X f(x) = \int_X \langle \nabla_X f(x), \boldsymbol{v}(x) \rangle_{T_x X} \, dx$$

whereby $\boldsymbol{v} : X \to TX$ represents a vector field and $\langle \cdot, \cdot \rangle_{T_x X} : T_x X \times T_x X \to \mathbb{R}$ the Riemannian metric on $X$.

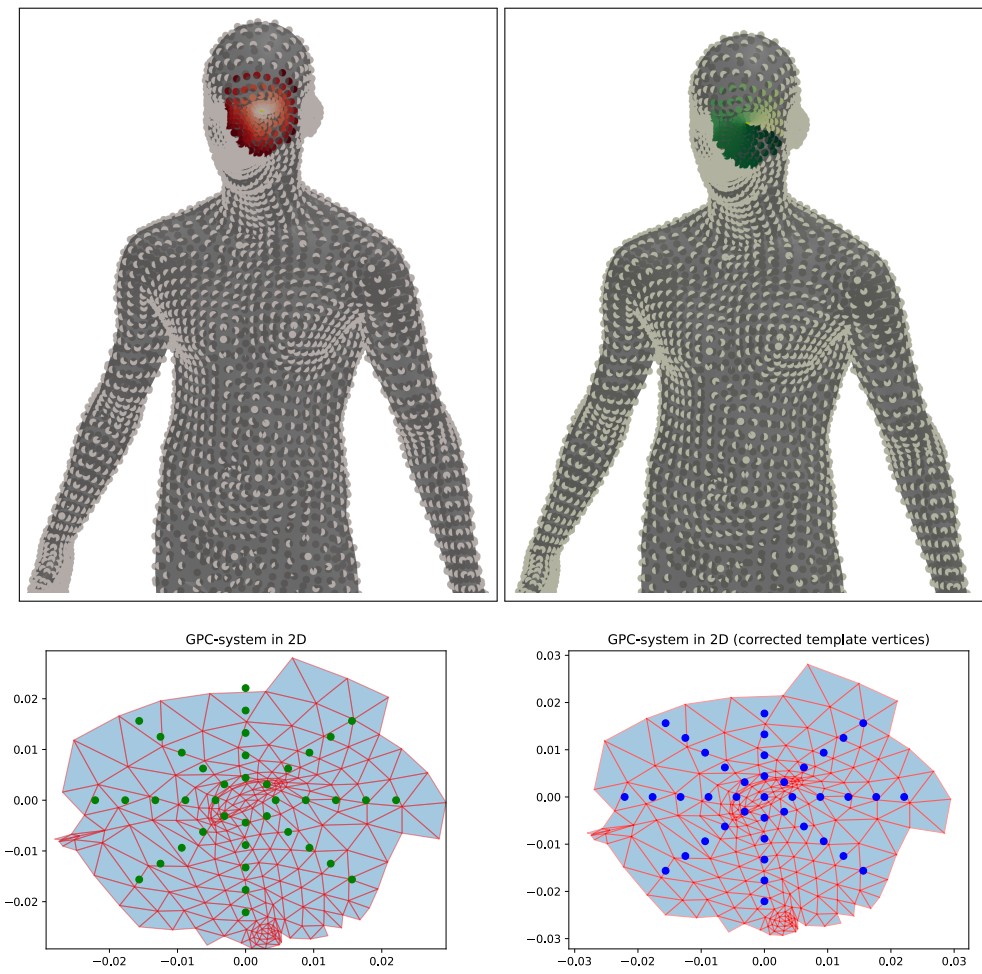

Figure 4: A visualization of a GPC-system with the center in a mesh vertex $\boldsymbol{v}_k$ that is roughly positioned in the eye of a mesh from the FAUST-dataset Bogo et al. (2014) and the discretized template. **[Top Left]** The *radial coordinates* of the vertices captured by the local GPC-system. Darker red colors indicate larger distances to the center vertex. The GPC-system has a maximum geodesic radius of $R \approx 0.03$ **[Top Right]** The *angular coordinates* of the vertices captured by a local GPC-system. Darker green colors indicate larger angular distances to the reference direction. **[Bottom Left]** The GPC-system from the mesh in the 2D-plane. The green vertices represent all template vertices $\mathbb{T}$ for $N_\rho = 5$ and $N_\theta = 8$ and $\rho_0 \approx 0.02$. Note that in practice it can happen, that a template vertex lies in no triangle. We circumvent that issue as described in the next picture. **[Bottom Right]** The same GPC-system in the 2D-plane. Here, however, we have corrected template vertices. All template vertices which do not fall into any triangle captured by the underlying GPC-system, receive barycentric coordinates of $\boldsymbol{0}$ and thus contribute a signal of $\boldsymbol{0}$ to the convolution. Visually, they will lie in the center due to the barycentric coordinates being $\boldsymbol{0}$. We compute the convolution for the center vertex $\boldsymbol{v}_k$, by summing over the matrix-vector-products of the (trainable) weight matrices $\mathsf{T}$ and the signal-vectors at the template vertices $(\rightarrow (\boldsymbol{S} * \mathsf{T})_\mathsf{W}(\boldsymbol{v}_k))$. The signal at each template vertex $(\rho_r, \theta_a)$ contains prior knowledge $\mathsf{W}_{ra}$ $(\rightarrow [D_\mathsf{W}(\boldsymbol{v}_k)\boldsymbol{S}](\rho_r, \theta_a))$. The "raw" signal under each template vertex $(\rho_x, \theta_y)$ is determined by the values of their surrounding triangle vertices $\left(\rightarrow \sum_{i=0}^2 \mathsf{B}_{kxyi1} \boldsymbol{S}_{\mathsf{B}_{kxyi0}}\right)$.

# B  PROOFS

*Proof.* Proof for Theorem 1: The parametric patch operator of Monti et al. (2017) for continuous manifolds is given by:

$$[D_p(\boldsymbol{u})s](\boldsymbol{v}) = \int_{B_R(\boldsymbol{0})} p_{\boldsymbol{v}}(\boldsymbol{y})[s \circ \exp_{\boldsymbol{u}} \circ \omega_{\boldsymbol{u}}](\boldsymbol{y}) \, d\boldsymbol{y}$$

Inserting it into the definition for intrinsic surface convolutions by Bronstein et al. (2021) yields:

$$
\begin{aligned}
(s * t)_{\Delta\theta,p}(\boldsymbol{u}) &= \int_{B_R(\boldsymbol{0})} t(\boldsymbol{v})[D_p(\boldsymbol{u})s](\boldsymbol{v}) \, d\boldsymbol{v} \\
&= \int_{B_R(\boldsymbol{0})} t(\boldsymbol{v}) \int_{B_R(\boldsymbol{0})} p_{\boldsymbol{v}}(\boldsymbol{y})[s \circ \exp_{\boldsymbol{u}} \circ \omega_{\boldsymbol{u}}](\boldsymbol{y}) \, d\boldsymbol{y}d\boldsymbol{v} \\
&= \int_{B_R(\boldsymbol{0})}\int_{B_R(\boldsymbol{0})} t(\boldsymbol{v})p_{\boldsymbol{v}}(\boldsymbol{y})[s \circ \exp_{\boldsymbol{u}} \circ \omega_{\boldsymbol{u}}](\boldsymbol{y}) \, d\boldsymbol{y}d\boldsymbol{v}
\end{aligned}
$$

Since $[s \circ \exp_{\boldsymbol{u}} \circ \omega_{\boldsymbol{u}}]$ is continuous as a composition of continuous functions and $t, p_{\boldsymbol{v}} \in C^0(\mathbb{R}^n)$ by assumption, we can apply the theorem of Fubini to change the order of the integrals:

$$
\begin{aligned}
&\int_{B_R(\boldsymbol{0})}\int_{B_R(\boldsymbol{0})} t(\boldsymbol{v})p_{\boldsymbol{v}}(\boldsymbol{y})[s \circ \exp_{\boldsymbol{u}} \circ \omega_{\boldsymbol{u}}](\boldsymbol{y}) \, d\boldsymbol{v}d\boldsymbol{y} \\
&= \int_{B_R(\boldsymbol{0})} \left[ \int_{B_R(\boldsymbol{0})} t(\boldsymbol{v})p_{\boldsymbol{v}}(\boldsymbol{y}) \, d\boldsymbol{v} \right] [s \circ \exp_{\boldsymbol{u}} \circ \omega_{\boldsymbol{u}}](\boldsymbol{y}) \, d\boldsymbol{y} \\
&= \int_{B_R(\boldsymbol{0})} \widetilde{p}_t(\boldsymbol{y})[s \circ \exp_{\boldsymbol{u}} \circ \omega_{\boldsymbol{u}}](\boldsymbol{y}) \, d\boldsymbol{y} \\
&= (s * \widetilde{p}_t)_{\Delta\theta}(\boldsymbol{u})
\end{aligned}
$$

$\square$

# C  PRIORS

- **Dirac Prior:**
$$\delta_{\overline{\rho},\overline{\theta}}(\rho,\theta) = \begin{cases} 1 & \text{if } \rho = \overline{\rho} \text{ and } \theta = \overline{\theta} \\ 0 & \text{else.} \end{cases}$$

- **Exponential Prior:**
$$\varepsilon_{\overline{\rho},\overline{\theta}}(\rho,\theta) = \lambda^2 \exp\left(-\lambda\left(d_r(\rho,\overline{\rho}) + d_a(\theta,\overline{\theta})\right)\right)$$

- **$\mathcal{X}^2$ Prior:**
$$\mathcal{X}^2_{\overline{\rho},\overline{\theta}}(\rho,\theta) = \left(\frac{1}{2^{n/2}\,\Gamma(n/2)}\right)^2 d_r(\rho,\overline{\rho})^{\frac{n}{2}-1} d_a(\theta,\overline{\theta})^{\frac{n}{2}-1} \exp\left(-\frac{d_r(\rho,\overline{\rho}) + d_a(\theta,\overline{\theta})}{2}\right)$$
with
$$\Gamma(1/2) = \sqrt{\pi}, \ \Gamma(1) = 1, \ \Gamma(r+1) = r \cdot \Gamma(r)$$

- **Student-t Prior:**
$$s_{\overline{\rho},\overline{\theta}}(\rho,\theta) = \left(\frac{\Gamma((n+1)/2)}{\sqrt{\pi n}\,\Gamma(n/2)}\right)^2 \left(1 + \frac{d_r(\rho,\overline{\rho})^2}{n}\right)^{-\frac{n+1}{2}} \left(1 + \frac{d_a(\theta,\overline{\theta})^2}{n}\right)^{-\frac{n+1}{2}}$$
with
$$\Gamma(n+1) = n!, \ \Gamma\left(n+\frac{1}{2}\right) = \frac{(2n)!}{n!4^n}\sqrt{\pi}$$

Thereby, the distance metrics in the radial coordinates $d_r(\cdot, \cdot)$ and angular coordinates $d_a(\cdot, \cdot)$ are given by:

$$d_r(\rho, \overline{\rho}) = |\rho - \overline{\rho}|$$

$$d_a(\theta, \overline{\theta}) = \begin{cases} \min\{\theta - \overline{\theta}, \overline{\theta} + 2 * \pi - \theta\} & \text{if } \theta \geq \overline{\theta} \\ \min\{\overline{\theta} - \theta, \theta + 2 * \pi - \overline{\theta}\} & \text{else.} \end{cases}$$

