# OpenReview forum: "Intrinsic Mesh CNNs"
_ICLR.cc/2024/Conference — Submitted to ICLR 2024_

### Official Review · Reviewer_jqVq · 2023-10-30

**Soundness:** 2 fair
**Presentation:** 3 good
**Contribution:** 1 poor
**Rating:** 3
**Confidence:** 4

**Summary:**

In this paper, the authors relate three special cases of intrinsic convolutions on surfaces to the generic formulation in terms of gauge and template functions.

They further show how different choices of templates can influence the features learnt in conjunction with the weighting function, naming these choices prior as they may encode assumptions made about the surface properties the network can learn.

From the formulation of Monti, they introduce a Dirac distribution prior taken as the zero variance limit of Gaussian functions. They characterise the set of learnable features for the Dirac prior and claim it is more general than others as it is "not limited by an integral".

Finally, the authors conduct experimental evaluations using different choices of priors, taken to be the density functions of several usual probability distributions. In the experimental setting of learning dense point correspondences, they show the Dirac prior outperforms the other choices.

**Strengths:**

The paper relates three milestone formulations of intrinsic convolutions on surfaces to the general theory of convolution in the tangent space with a choice of gauge. The authors introduce a partial order to the set of priors through a notion of what prior is more powerful - i.e., can a prior learn the same features as another.

**Weaknesses:**

The authors focus on a limited body of work in the geometric deep learning literature, namely, three early formulations of intrinsic convolutions. Furthermore, they only cite one reference regarding all the theoretical framework of gauge equivariant CNNs, Bronstein et al., 2021. Combined, these two factors mean the paper ignores important references on gauge equivariant CNNs published before (Cohen et al.) or at the same time (Weiler et al.) the book cited, as well as other related works on convolutions for surfaces and 3D shapes (e.g., Tangent Convolutions for Dense Prediction in 3D in CVPR 2018, MeshCNN, and others).

The mathematical derivations contain typos (the limit of Gaussian functions for decreasing variance should be for n -> 0 not n -> +oo), and the use of the notation $\Delta_{\theta}$ and $\Delta_{\theta, w}$ should be introduced. The authors should introduce the change of integration domain from [0, 1]^n to BR(0) before Theorem 1. The theorem itself is a direct application of Fubini's theorem and as such is not a particularly strong theoretical result of the paper.

As noted by the authors themselves, the Dirac distribution is not a function. This weakens the link with the theoretical results presented before.

The authors claim to "we show that existing definitions for surface convolutions only differ in their prior assumptions about local surface information". It is known in the community, and the authors show it for the formulation of Monti only, who had already shown their formulation encompasses the previous Geodesic CNN and Anisotropic CNN.

Finally, the experimental evaluation yields important questions (detailed below).

**Questions:**

Can the theoretical framework be strengthened by reformulating it in terms of distributions and extended to the class of test functions?

The network architecture used in the experiments uses angular max pooling. Can the authors clarify why this is needed?
Isn't it a step backwards compared to the work of Monti, or other mesh CNNs such as FeastNet that do not require angular max pooling?

How did the authors choose the priors compared against the Dirac prior?

---

> ### Author Response · Authors · 2023-11-21
>
> We are very grateful for the provided review. In the following, we would like to address the points and questions.
>
> > The authors focus on a limited body of work in the geometric deep learning literature, namely, three early formulations of intrinsic convolutions. Furthermore, they only cite one reference regarding all the theoretical framework of gauge equivariant CNNs, Bronstein et al., 2021.
>
> The theoretical contribution of our work focuses on the discovery of previously unknown but insightful
> properties of intrinsic surface convolutions which depend only on the definition of priors and is as such
> more fundamental and independent of gauge transformations. Therefore, while we think our theory
> also applies to gauge-equivariant intrinsic surface convolutions, this topic exceeds the scope of this
> work.
>
> > The mathematical derivations contain typos (the limit of Gaussian functions for decreasing variance should be for $n \to 0$ not $n \to \infty$, and the use of the notation $\Delta_\theta$ and $\Delta_{\theta, w}$ should be introduced.
>
> We thank the reviewer for pointing out typos in our manuscript and will of course correct those, as well as add
> the full definition for the anisotropic Laplacian $\Delta_{\alpha \theta}$ into the appendix to ensure completeness.
>
> > As noted by the authors themselves, the Dirac distribution is not a function. This weakens the link with the theoretical results presented before.
>
> While mathematically possible to expand the domain of priors to transition kernels and with that
> allowing for several new priors such as the Dirac prior, this redefines the Riemann integrals into
> Lebesgue integrals which come with a significant amount of technical overhead, as stated in the manuscript,
> and whose description would not help to understand our contribution.
> Thereby, our choice of describing the Dirac prior allows to compare it to all previous models that use
> priors, which can be expressed in terms of Riemann integrals, for example by comparing their sets of
> learnable features. This, in turn, again shows the potential of reformulating IMCNNs in terms of using a
> prior rather than a patch operator.
>
> > The authors claim to "we show that existing definitions for surface convolutions only differ in their prior assumptions about local surface information". It is known in the community, and the authors show it for the formulation of Monti only, who had already shown their formulation encompasses the previous Geodesic CNN and Anisotropic CNN.
>
> We generalize the framework of Monti et al., 2017, thereby obviously allowing for the definition of
> models mentioned by Masci et al., 2015, and Boscaini et al., 2016, but additionally also the model
> of Bronstein et al., 2021, and new models that use priors in the sense Theorem 1.
>
> > Can the theoretical framework be strengthened by reformulating it in terms of distributions and extended
> to the class of test functions?
>
> While the discussion about distributions and test functions definitely is possible, we are convinced
> that the choice of transition kernels is more natural as we think of signals on the manifold in a
> statistical fashion. If we assume that our IMCNNs only learn test functions, we would restrict our
> research on a comparably limited domain.
>
> > The network architecture used in the experiments uses angular max pooling. Can the authors clarify
> why this is needed? Isn’t it a step backwards compared to the work of Monti, or other mesh CNNs such
> as FeastNet that do not require angular max pooling?
>
> Angular max-pooling (AMP) is required as a means to resolve the angular coordinate ambiguity
> problem, a direct consequence of gauge-ambiguity, in practice. It was also used by Monti et al., 2017.
> FeastNet (Verma et al., 2018), while representing interesting work, is a fundamentally different
> approach to surface convolutions.
>
> >  How did the authors choose the priors compared against the Dirac prior?
>
> The normal distribution has also been used by Monti et al., 2017. The other priors portray valid instances
> for our class of IMCNNs and were selected for further comparison.
>
> **Sources**
> > *Masci, Jonathan, et al. "Geodesic convolutional neural networks on riemannian manifolds." Proceedings of the IEEE international conference on computer vision workshops. 2015.*
>
> > *Boscaini, Davide, et al. "Learning shape correspondence with anisotropic convolutional neural networks." Advances in neural information processing systems 29 (2016).*
>
> > *Monti, Federico, et al. "Geometric deep learning on graphs and manifolds using mixture model cnns." Proceedings of the IEEE conference on computer vision and pattern recognition. 2017.*
>
> > *Verma, Nitika, Edmond Boyer, and Jakob Verbeek. "Feastnet: Feature-steered graph convolutions for 3d shape analysis." Proceedings of the IEEE conference on computer vision and pattern recognition. 2018.*
>
> > *Bronstein, Michael M., et al. "Geometric deep learning: Grids, groups, graphs, geodesics, and gauges." arXiv preprint arXiv:2104.13478 (2021).*

---

### Official Review · Reviewer_eZPj · 2023-11-01

**Soundness:** 2 fair
**Presentation:** 3 good
**Contribution:** 2 fair
**Rating:** 3
**Confidence:** 3

**Summary:**

This paper proposes unifying the intrinsic spatial convolution on manifolds from Bronstein et al 2021 with the spatial convolution and patch operators in Monti et al 2017. The authors show that there is an implicit prior by connecting the two formulations and propose a class of intrinsic CNNs with different priors.

**Strengths:**

The authors describe previous work (Masci et al 2015, Boscaini et al 2016, Monti et al 2017) in detail and provide a good background of the different patch operators.

**Weaknesses:**

-My primary concern with this paper is that it’s not clear to me why it is called a prior and how using a different prior helps. It simply seems like different instances of the patch operator, akin to the Gaussian in Masci et al. 2015.

-If the Dirac prior is the most expressive, why should one consider other priors? The results don’t seem to show much difference between the other priors.

-Are the shot descriptors orientation invariant? Does a global gauge exist?

-Why was a gauge-equivariant method such as [1] not considered? It seems very relevant to the proposed method and has the advantage that the output transforms accordingly with the gauge transformations.

References:
[1] De Haan, P., Weiler, M., Cohen, T., & Welling, M. (2020, October). Gauge Equivariant Mesh CNNs: Anisotropic convolutions on geometric graphs. In International Conference on Learning Representations.

**Questions:**

See weaknesses.

---

> ### Author Response · Authors · 2023-11-21
>
> We appreciate the provided feedback of the reviewer and would like to address the raised points.
>
> > My primary concern with this paper is that it’s not clear to me why it is called a prior and how using a different prior helps. It simply seems like different instances of the patch operator, akin to the Gaussian in Masci et al. 2015.
>
> The prior can be seen as a formal tool for encoding assumptions that you have about the signal on the surface.
> Our contribution formally describes the effects of priors. Key to the description is the introduction
> of the notion of *learnable features* in Section 4 of our manuscript. Depending on the chosen prior, the set of learnable
> features of an IMCNN can be more or less comprehensive. Given a problem, like for example classification, we search for
> features within the set of learnable features that cause the IMCNN to perform well. If the set of learnable
> features is smaller, we expect that good features within that set are found faster.
> However, a small set of learnable features might exclude features which lead to better performances.
> Larger sets of learnable features, in turn, imply more unstable training as searching for good features
> becomes more complicated due to the increased amount of possible features to learn.
>
> > If the Dirac prior is the most expressive, why should one consider other priors? The results don’t seem to show much difference between the other priors.
>
> Our paper introduced the Dirac prior as a means to connect the framework of Monti et al., 2017, with the definition of intrinsic surface convolutions from Bronstein et al., 2021. We subsequently discovered that the Dirac prior is very comprehensive as it is associated to a large set of learnable features. This comes with a higher probability of finding more suitable features for the underlying problem, as indicated by the experiment results in which Dirac outperforms the other priors. However, this comes at the cost of a large search space. If unstable training is observed and domain knowledge is available, we have the option to limit our network onto a certain set of learnable features to stabilize training by using a more restrictive prior.
>
> > Are the shot descriptors orientation invariant?
>
> SHOT-descriptors are rotation-invariant as they are computed with the help of relative angles between
> vertex normals, which do not vary under the application of isometries. However, our work
> does not discuss SHOT-descriptors. Therefore, we kindly refer to the original paper (Salti et al.,
> 2014) for further questions about SHOT-descriptors.
>
> > Does a global gauge exist?
>
> A global gauge does not exist in the general case. A gauge $\omega$ defines a local basis for a tangent
> space $T_u M$ with $u \in M$ of the manifold $M$. In the general case, $T_u M$ and $T_v M$ are different vector spaces for
> $u \not = v$, which already shows that a gauge usually cannot be globally equal for multiple tangent spaces.
> However, in the special case that the underlying manifold is parallelizable, we at least have a global
> continuously changing gauge.
>
> > Why was a gauge-equivariant method such as [1] not considered? It seems very relevant to the proposed method and has the advantage that the output transforms accordingly with the gauge transformations.
>
> The theoretical contribution of our work focuses on the discovery of previously unknown but insightful
> properties of intrinsic surface convolutions which depend only on the definition of priors and is as
> such more fundamental and independent from gauge transformations. Therefore, while we think our
> theory also applies to gauge-equivariant intrinsic surface convolutions, this topic exceeds the scope
> of this work.
>
> **Sources**
> > *Salti, Samuele, Federico Tombari, and Luigi Di Stefano. "SHOT: Unique signatures of histograms for surface and texture description." Computer Vision and Image Understanding 125 (2014): 251-264.*
>
> > *Monti, Federico, et al. "Geometric deep learning on graphs and manifolds using mixture model cnns." Proceedings of the IEEE conference on computer vision and pattern recognition. 2017.*
>
> > *Bronstein, Michael M., et al. "Geometric deep learning: Grids, groups, graphs, geodesics, and gauges." arXiv preprint arXiv:2104.13478 (2021).*

---

> > ### Comment · Reviewer_eZPj · 2023-11-23
> >
> > I thank the authors for the detailed response. I'm afraid I still do not see the value of this work. I still do not understand how the use of priors is important for understanding intrinsic surface convolutions and the experiments do not seem to elucidate any additional insights. I therefore maintain my original score.

---

### Official Review · Reviewer_u4PC · 2023-11-06

**Soundness:** 1 poor
**Presentation:** 2 fair
**Contribution:** 1 poor
**Rating:** 3
**Confidence:** 4

**Summary:**

The authors attempt to make new connections between prior works on convolutions on meshes. They explore a novel parametrization of the convolution operation, and find that it performs worse than the typical parametrization.

**Strengths:**

- I appreciate the attempt of drawing new connections between prior works.

**Weaknesses:**

I am afraid I do not understand the point of this work.

Any linear map from a scalar signal $s: M \to \mathbb R$ on a $n$-manifold, to another scalar signal can be represented by a function $w : M \times M \to \mathbb R$, via the integral $(s \star w)(x) = \int\_M w(x, y)s(y)dy$. Typically, for a convolution-style operator, one does not want the parameters to depend on $x$, so one chooses a frame/gauge $\omega$ on the manifold and arrives at the "intrinsic manifold convolution" of def 1 of the manuscript, with a parameter $t: \mathbb R^n \to \mathbb R$.

Monti et al (2017) [4] recognize that the gauge can not be chosen uniquely, so choose $J$ frames apply a convolution on each of choice of frame, with different parameters each. In other words, they choose a parameter function $t : \mathbb R^n \to \mathbb R^J$.
The alternative approach suggested by [2,3] is to be equivariant to the choice of frame, leading to constraints on the parameters (and typically the use of non-scalar features such as vectors).

However, what's done in theorem 1 in the equation of $(s \star t)\_{\Delta \theta, w}(u)$, is very different from any of the above. Instead of computing the output signal at one point by a single integral (over either the manifold or over the tangent plane), the authors compute a convolution as two integrals over the tangent plane.
Also, they use one parameterization $\mathbb R^n \to \mathbb R$, and another parametrization $\mathbb R^n \times \mathbb R^n \to \mathbb R$.

The authors appear to suggest that this is similar to what Monti et al (2017) [4] does, but this appears to me as very different. As the authors themselves note in theorem 1, this notation is completely redundant and can be reduced to a single integral. In fact, the equation for $(s \star t)\_{\Delta \theta, w}(u)$ appears to do two convolutions, with different parameters, which should indeed reduce to a single convolution. In the rest of sec 3, the authors make the unsurprising observation that if one of the two convolutions contain a Dirac delta, that convolution is an identity operation, and the double convolution reduces to just the second convolution.

So what's the point of analyzing the double convolution, which no one uses? It's very different from what [4] proposes, so how does it bridge a gap between anything?

In section 5, the authors are considering parametrizations of the double convolution different from the Dirac prior (thus the single convolution) and find that they perform worse. Also, as they involve a double integral, I suspect that they are much slower to compute.

In short, the authors didn't make any new connection between prior works, and proposed a different parametrization of the convolution that performed worse. In case I completely misunderstood the work, I look forward to your clarifications and will reconsider my opinion.

Other points:
- The authors cite [1] for the gauge equivariant convolution. This should be a citation to [2] and also [3] in the context of meshes.
- The authors should compare to [3] in their experiments. Those authors found close to 100% correspondence at 0 geodesic distance for the the FAUST experiment, much better than the numbers reported in the present manuscript.
- It's very confusing to use $w$ (w) and $\omega$ (omega) in the same equation, the first referring to weights and the latter to the gauge. Please choose an alternative notation.


Refs:
- [1] Bronstein, Michael M., Joan Bruna, Taco Cohen, and Petar Veličković. 2021. “Geometric Deep Learning: Grids, Groups, Graphs, Geodesics, and Gauges.” http://arxiv.org/abs/2104.13478.
- [2] Cohen, Taco S., Maurice Weiler, Berkay Kicanaoglu, and Max Welling. 2019. “Gauge Equivariant Convolutional Networks and the Icosahedral CNN.” http://arxiv.org/abs/1902.04615.
- [3] De Haan, P., M. Weiler, T. Cohen, and M. Welling. 2020. “Gauge Equivariant Mesh CNNs: Anisotropic Convolutions on Geometric Graphs.” https://arxiv.org/abs/2003.05425.
- [4] Monti, Federico, Davide Boscaini, Jonathan Masci, Emanuele Rodolà, Jan Svoboda, and Michael M. Bronstein. 2016. “Geometric Deep Learning on Graphs and Manifolds Using Mixture Model CNNs.” http://arxiv.org/abs/1611.08402.

**Questions:**

see above

---

> ### Author Response · Authors · 2023-11-21
>
> We thank the reviewer for the undertaken efforts and want to address the raised concerns.
>
> > Any linear map $s: M \to \mathbb{R}$ from a scalar signal on a $n$-manifold, to another scalar signal can be represented by a function [...]
>
> The formulation of the intrinsic manifold convolution follows a different intention. Intrinsic manifold
> convolutions (Bronstein et al., 2021) represent a generalization from standard Euclidean convolutions
> to convolutions on Riemannian manifolds. For this we require a notion of relative direction between
> points within the manifold. Bronstein et al., 2021, proposes to use tangent vectors. We
> associate tangent vectors to coordinates by selecting one from multiple possible gauges for each
> tangent plane. Eventually, we map said tangent vectors via the exponential map
> onto the surface to retrieve manifold points on which the surface signal is defined.
>
> > Monti et al (2017) [4] recognize that the gauge can not be chosen uniquely, so choose $J$ frames apply [...]
>
> Monti et al., 2017, proposes a general algorithmic framework for the construction of previous geometric CNN
> architectures such as, but not limited to, geodesic CNNs (Masci et al., 2015). Within said framework, when
> specifying for the geodesic CNNs, $J$ refers to the amount of neighbors around surface point $x$,
> at which we calculate the convolution. In the continuous case, the sum becomes an integral over the disc
> $B_R(\textbf{0})$, that contains geodesic polar coordinates which portray the input to the selected gauge.
> Thereby, $R$ is pre-specified radius that typically refers to the injectivity radius of the earlier discussed
> exponential map.
>
> > However, what's done in theorem 1 in the equation of, is very different from any of the above. [...]
>
> Our convolution is equivalent to how Monti et al., 2017, and Bronstein et al., 2021, would compute
> intrinsic surface convolutions. We have shown this in Theorem 1. The proof is provided in the appendix
> and comes down to the application of Fubini’s theorem. The function $t : \mathbb{R}^n \to \mathbb{R}$
> represents the template that shall be learned, the function $w : \mathbb{R}^n \times \mathbb{R}^n \to \mathbb{R}$
> references a hand-crafted prior.
>
> > The authors appear to suggest that this is similar to what Monti et al (2017) [4] does, but this appears to me as very different. [...] So what's the point of analyzing the double convolution, which no one uses? It's very different from what [4] proposes, so how does it bridge a gap between anything?
>
> As noted before, Theorem 1 states that the framework of Monti et al., 2017, is equivalent to our
> notation. Selecting Dirac as a prior does not make the intrinsic surface convolution an identity
> operation, but returns the formulation for intrinsic surface convolutions of Bronstein et al., 2021.
> That is, selecting Dirac as a prior within our reformulation of Monti et al., 2017, which we have
> proven to be equivalent, closes the theoretical gap between the algorithmic framework of
> Monti et al., 2017, and the theoretical derivation of the intrinsic surface convolution from
> Bronstein et al., 2021.
>
> > The authors cite [1] for the gauge equivariant convolution. This should be a citation to [2] and also [3]
> in the context of meshes.
>
> We are aware of the literature by Taco et al., 2019, and De Haan et al., 2020. We thank the reviewer
> for pointing out that these citations are missing and will of course add them to the manuscript.
>
> > The authors should compare to [3] in their experiments. Those authors found close to 100 correspondence
> at 0 geodesic distance for the the FAUST experiment [...]
>
> The theoretical contribution of our work focuses on the discovery of previously unknown but insightful
> properties of intrinsic surface convolutions and not on extensions of them such as gauge-equivariant
> convolutions.
>
> > It’s very confusing to use $w$ (w) and $\omega$ (omega) in the same equation, the first referring to weights and the
> latter to the gauge.
>
> Thank you for your feedback on the readability of our paper. We can rephrase $w(·)$ to $p(·)$,
> highlighting that $w(·)$ does not contain trainable weights but represents a handcrafted prior.
>
> **Sources:**
> > *Masci, Jonathan, et al. "Geodesic convolutional neural networks on riemannian manifolds." Proceedings of the IEEE international conference on computer vision workshops. 2015.*
>
> > *Monti, Federico, et al. "Geometric deep learning on graphs and manifolds using mixture model cnns." Proceedings of the IEEE conference on computer vision and pattern recognition. 2017.*
>
> > *Cohen, Taco, et al. "Gauge equivariant convolutional networks and the icosahedral CNN." International conference on Machine learning. PMLR, 2019.*
>
> > *De Haan, Pim, et al. "Gauge equivariant mesh CNNs: Anisotropic convolutions on geometric graphs." arXiv preprint arXiv:2003.05425 (2020).*
>
> > *Bronstein, Michael M., et al. "Geometric deep learning: Grids, groups, graphs, geodesics, and gauges." arXiv preprint arXiv:2104.13478 (2021).*

---

> ### Comment · Reviewer_u4PC · 2023-11-21
>
> Dear authors,
>
> I thank you for your reply.
>
> Can you refer exactly to where Monti [4] refers to $J$ as the number of neighbours, in the context of their proposed approach? Under equation (9) in [4], they write
>
> > $J$ represents the dimensionality of the extracted patch
>
> Later they write that there are $2Jd$ parameters. The index $J$ is thus a statement about their parametrization and has nothing to do with the number of neighbours. In fact, it's the number of mixture components in their CNN, thus it's a discrete index. I therefore fail to see why it makes sense to take that index to have values in the tangent space, and also fail to see why your description theorem 1 has anything to do with Monti's formulation.
>
> Could you please elaborate on why your formulation in theorem 1 is a sensible "continuous version" of Monti [4]?
>
> [4] Monti, Federico, Davide Boscaini, Jonathan Masci, Emanuele Rodolà, Jan Svoboda, and Michael M. Bronstein. 2016. “Geometric Deep Learning on Graphs and Manifolds Using Mixture Model CNNs.” http://arxiv.org/abs/1611.08402.

---

> > ### Author Response · Authors · 2023-11-22
> >
> > Dear reviewer,
> > > Can you refer exactly to where Monti [4] refers to $J$ as the number of neighbours, in the context of their proposed approach? Under equation (9) in [4], they write "$J$ represents the dimensionality of the extracted patch". Later they write that there are $2Jd$ parameters. The index $J$ is thus a statement about their parametrization and has nothing to do with the number of neighbours. In fact, it's the number of mixture components in their CNN, thus it's a discrete index. I therefore fail to see why it makes sense to take that index to have values in the tangent space, and also fail to see why your description theorem 1 has anything to do with Monti's formulation.
> >
> > A convolution in a point $x \in M$, where $M$ represents the underlying manifold, represents a local operation as it aggregates over a local neighborhood.
> > In the context of parameterizing the framework from Monti et al., 2017, for geodesic CNNs (Masci et al., 2015), we select neighbors for the convolution by selecting mean values $\bar{u}_j$ for Gaussians $w_j(u)$ which are defined over geodesic polar coordinates.
> > The signal value at neighbor $\bar{u}_j$ is an interpolated signal (Masci et al., 2015; Monti et al., 2017) that uses the values of the Gaussian $w_j(u)$ as interpolation values (Equation 9 in Monti et al., 2017):
> >
> > $$
> > D_j(x)f = \sum_{y \in \mathcal{N}(x)} w_j(u(x, y)) f(y),\ j=1,...,J
> > $$
> >
> > The extracted patch at $x \in M$ captures the locally extracted signal in a vector $\vec{p}_f(x) = [D_1(x)f, D_2(x)f, ..., D_J(x)f]$.
> > Thereby, each entry corresponds to the interpolated signal at one neighbor $\bar{u}_j$.
> > In total we have $J$ neighbors, causing the extracted patch $\vec{p}_f(x)$ to have a dimensionality of $J$.
> > Eventually, Monti et al., 2017, phrase the convolution in $x \in M$ as an inner product between trainable weights $\vec{g}$ and the
> > extracted patch at $x$ in Equation 10:
> >
> > $$
> > (f \ast \vec{g})(x) = \sum_{j=1}^J g_j D_j(x)f = \vec{g}^T \vec{p}_f(x)
> > $$
> >
> > > Could you please elaborate on why your formulation in theorem 1 is a sensible "continuous version" of Monti [4]?
> >
> > Since our theory considers Riemannian manifolds, Theorem 1 uses integrals instead of sums, as it has also been suggested by Monti et al., 2017. Furthermore, since we now have specified our data domain, we can be more elaborate on how we read signals from our data. This allows for a specification of $f(y)$ from Equation 9 in Monti et al., 2017. This specification is not only discussed in the background section of our manuscript, but also in excellent detail in Bronstein et al., 2021. The exponential map, which is included in the specification, portrays a local diffeomorphism that is limited by the so called injectivity radius $R$. Therefore, $B_R(\textbf{0})$ yields a valid integration domain.
> >
> > We hope we could clarify any misunderstandings.
> >
> > **Sources**:
> > > *Masci, Jonathan, et al. "Geodesic convolutional neural networks on riemannian manifolds." Proceedings of the IEEE international conference on computer vision workshops. 2015.*
> >
> > > *Monti, Federico, et al. "Geometric deep learning on graphs and manifolds using mixture model cnns." Proceedings of the IEEE conference on computer vision and pattern recognition. 2017.*
> >
> > > *Bronstein, Michael M., et al. "Geometric deep learning: Grids, groups, graphs, geodesics, and gauges." arXiv preprint arXiv:2104.13478 (2021).*

---

> > > ### Comment · Reviewer_u4PC · 2023-11-22
> > >
> > > Dear authors,
> > >
> > > I guess we have a very different notion of the word "neighbour". In my terminology, the set of neighbours of a point $x$, is the set $\mathcal N(x)$, of points $y$ that are close to / connected to point $x$ on the manifold. You're also using this terminology in section 2.3.
> > >
> > > In contrast, what you call "neighbour" in this discussion I'd call "patch", namely a region of the tangent space that we want to apply the same weight to. The operator $D_j(x)f$ computes the value of the $j$'th patch at position $x$ of signal $f$. Subsequently, the weight vector $g$ weighs the different patches.
> > >
> > > Monti [4] considers a finite number of patches, $J$. In my eyes, it's very different to take the space of patches to be uncountably infinite and parametrize it by the tangent space. While I understand your construction, I fail to see why this is the obvious "the continuous version of the parametric patch operator from Monti", as you write in theorem 1. As such, I fail to see why this draws a meaningful connection to the Gaussian mixture CNN proposed by Monti [4], and the generic intrinsic mesh convolution.
> > >
> > > The only meaningful connection I can see it the opposite from the one you draw: to interpret the Gaussian mixture CNN as an instance of the intrinsic mesh CNN. If the pseudo-coordinates $u(x,y)$ are Riemannian normal coordinates, simply set the intrinsic mesh CNN template to be the mixture of Gaussians: $t(v)=\sum_{j=1}^J g_j w_j(v)$.

---

> > > > ### Author Response · Authors · 2023-11-22
> > > >
> > > > > I guess we have a very different notion of the word "neighbour". In my terminology, the set of neighbours of a point $x$, is the set $\mathcal{N}(x)$, of points $y$ that are close to / connected to point $x$ on the manifold. You're also using this terminology in section 2.3.
> > > >
> > > > We refer to a point $y \in M$ as a neighbor of a point $x \in M$ if it is sufficiently close to $x$, i.e, within the injectivity radius $R$ of the exponential map.
> > > >
> > > > > In contrast, what you call "neighbour" in this discussion I'd call "patch", namely a region of the tangent space that we want to apply the same weight to. The operator $D_j(x)f$ computes the value of the $j$'th patch at position $x$ of signal $f$. Subsequently, the weight vector $g$ weighs the different patches.
> > > >
> > > > According to Monti et al., 2017, each $D_j(x)f$ is induced by a $w_j(u)$ which is a Gaussian, when specifying for geodesic CNNs (Masci et al., 2015), on the parameter space $B_R(\textbf{0})$ that is parameterizing the tangent space $T_x M$ at $x$ in geodesic polar coordinates (see Table 1 in Monti et al., 2017). Thereby, $R$ is the injectivity radius of the exponential map at $x$.
> > > > Each $w_j(u)$ is defined by a mean value $\bar{u}_j \in B_R(\textbf{0})$ and a standard deviation.
> > > > Considering one selected gauge for the tangent space, $\bar{u}_j$ corresponds uniquely to a point in the tangent space.
> > > > Thus, by further mapping said point onto the manifold using the exponential map at $x$, $\bar{u}_j$ corresponds uniquely to a neighbor of $x$ in the manifold in the sense of our definition given above.
> > > > Therefore, each $D_j(x)f$ corresponds to a neighboring point specified by $\bar{u}_j$.
> > > >
> > > > > Monti [4] considers a finite number of patches, $J$. In my eyes, it's very different to take the space of patches to be uncountably infinite and parametrize it by the tangent space. While I understand your construction, I fail to see why this is the obvious "the continuous version of the parametric patch operator from Monti", as you write in theorem 1. As such, I fail to see why this draws a meaningful connection to the Gaussian mixture CNN proposed by Monti [4], and the generic intrinsic mesh convolution.
> > > >
> > > > Because we consider infinitely many neighbors $\bar{u}_j$ we integrate over $B_R(\textbf{0})$.
> > > > Consequently, we learn a function $g(\cdot)$ that provides infinetly many weights $g_j$.
> > > > In our manuscript, we referred to that function as $t(\cdot)$.
> > > > In particular, in case we consider weights $g_j$ only at finitely many points $\bar{u}_1,\dots,\bar{u}_J$ the integral reduces to the sum in Equation 10 of Monti et al., 2017.

---

> > > > > ### Comment · Reviewer_u4PC · 2023-11-22
> > > > >
> > > > > I thank the authors for their response. I understand that if the patches have a "mean", then the patch can be identified with this mean point, and you could call the patch a neighbour. In your formulation in thm 1, however, you never required your patch $p_v(y)$ to be a Gaussian-like patch with mean $v$, so I don't see why this interpretation holds in general. As such, I still find it confusing to call the patch parametrized by $v$ a "neighbour".
> > > > >
> > > > > Also, my opinion is unchanged that that the authors' departure from a finite number of Gaussian patches (as used by Monti) to an infinite number of patches, makes for a wholly different architecture. I'm afraid I thus don't see this paper as proposing a meaningful connection between the convolution proposed by Monti and the intrinsic mesh convolution. My score remains unchanged.

---

### Official Review · Reviewer_jCDn · 2023-11-07

**Soundness:** 3 good
**Presentation:** 2 fair
**Contribution:** 2 fair
**Rating:** 3
**Confidence:** 2

**Summary:**

The paper builds a connection between a practical implementation of mesh CNN (Monti et al., 2017) to Intrinsic Mesh CNN (Bronstein et al., 2021). By defining a template, Intrinsic Mesh CNN reduces to the mesh CNN defined by Monti et al. with a Dirac prior. The paper later experiments with different choices of priors and show that Dirac prior leads to better results.

**Strengths:**

- The paper connects a practical model with the theoretical framework of Intrinsic convolution on meshes.
- The intuition behind the idea is straightforward and easy to understand.
- The paper proves that a partial order exists for comparing these priors.
- The paper shows some quantitative results to compare the difference between different priors.

**Weaknesses:**

- Although by defining priors the paper builds a connection between theory and practice, the resulted model is not that useful. In particular, the Dirac prior, which corresponds to Monti et al., 2017, is still the best solution at least in the experiments in the paper.
- Indeed, instead of decoupling $w_t(\cdot)$ (the  into $w(\cdot)$ (the prior) and $t(\cdot)$ (the template) and learning the template, one may simply treat $w_t(\cdot)$ as the learnable parameter. If we are allowed to discretize $w_t(\cdot)$ with sufficient amount of parameters, parametering $w_t$ is flexible enough. Making a (not quite accurate) analogy to regular Euclidean CNNs: it seems to me what the paper presents is to pre-convolve some handcrafted $H$ with the original convolutional kernel $K$, with $H$ being something very restrictive in the sense of both capacity and optimization. It therefore does not surprise me too much that the Dirac prior (equivalent to directly parametering $w_t$) is better than all other variants.
- Continuing the point above, I believe there is no reason not to learn the priors at the same time. And doing this may lead to some additional benefits in optimization.
- The presentation seems a bit messy in the experiment section. Many descriptions can be simplified: for instance, I do not think one needs to write down the exact formula of cross entropy (I guess it is well known by the majority of the audience).
- The benchmark seems to be quite toy. I feel that if the paper includes empirical results on the tasks presented by MeshCNN, it will be much more convincing.

[1] Hanocka, Rana, et al. "Meshcnn: a network with an edge." ACM Transactions on Graphics (ToG) 38.4 (2019): 1-12.

**Questions:**

- How much time does the propose implementation with non-Dirac priors consume, compared to Monti et al., 2017?
- How much difference does it make to change the hyperparameters, including the ones for the GPC coordinate systems and template discretization, in Table 1?

---

> ### Author Response · Authors · 2023-11-21
>
> We thank the reviewer for the valuable feedback. In the following, we would like to briefly address the raised points and questions.
>
> > Point 1:
>
> While it is true that earlier experiments report better training results, a direct comparison with
> these experiments is misleading as they applied post-processing methods to the network output,
> such as functional maps in Masci et al., 2015, and Boscaini et al., 2016, or intrinsic Bayesian filters
> in Monti et al., 2017. Furthermore, the Dirac prior is a proposal of our work, which is fundamental
> to establish a connection between the theory of Bronstein et al., 2021, to the practical framework of
> Monti et al, 2017. Notably, it was not discussed in Monti et al, 2017.
>
> > Point 2 and 3:
>
> Learning the parameters directly has been suggested by Bronstein et al., 2021. Pre-convolving the
> template $t(·)$ with a handcrafted kernel $w(·)$ has implicitly been done in what previously has been
> referred to as the patch operator (Masci et al., 2015; Monti et al, 2017). Our paper proposes a
> reformulation of the parametric framework from Monti et al, 2017, which stresses the theoretical
> implications of handcrafted kernels $w(·)$ by giving rise to the notion of learnable features. This
> notion not only explains why the handcrafted kernel encodes prior knowledge, which in previous
> work has merely been referred to as either interpolation coefficients, weighting function or kernel
> (Masci et al., 2015; Monti et al, 2017), but also allows us to observe a partial order on priors in
> the context of intrinsic mesh CNNs. Therefore, we now know that different priors can vary in their
> comprehensiveness. One very interesting example is given by our proposed Dirac prior, which on the
> one hand, when inserted into our reformulation, yields the theoretical definition for intrinsic surface
> convolutions of Bronstein et al., 2021, and on the other hand renders the pre-convolution irrelevant.
>
> > Point 4:
>
> In order to simplify the repeatability of our experiments, we want to be precise about our experiment
> setup.
>
> > Point 5:
>
> While we agree that there exist more interesting benchmarks, we want to clarify that our benchmark
> is no artificial toy benchmark. For instance, compared to Hanocka et al., 2019, our input meshes
> have 41328 edges. Their benchmark used considerable smaller meshes with roughly 750 edges for
> mesh classification and roughly 2250 edges for mesh segmentation. Additionally, in difference to
> Hanocka et al., 2019, we have trained our IMCNN to predict dense point-correspondences. Predicting
> dense point-correspondences is commonly regarded as a notoriously hard problem in the computer
> vision community (Van Kaick et al., 2011). Lastly, the benchmark is a common benchmark in the
> background papers (Masci et al., 2015; Boscaini et al., 2016; Monti et al, 2017) .
>
> > Question 1:
>
> The Dirac-prior effectively gets rid of a matrix multiplication, therefore requiring less computation
> than the IMCNNs which use a specific prior.
>
> > Question 2:
>
> In order to get an intuition for what difference the hyperparameter make, it is helpful to take a
> look on Figure 4 in the appendix. While it is difficult to quantify how much difference varying
> hyperparameter cause in general, we expect certain aspects to hold true. For instance, the GPC-
> system radius should be chosen smaller or equal to the injectvity radius of the exponential map.
>
> **Sources:**
>
> > *Van Kaick, Oliver, et al. "A survey on shape correspondence." Computer graphics forum. Vol. 30. No. 6. Oxford, UK: Blackwell Publishing Ltd, 2011.*
>
> > *Masci, Jonathan, et al. "Geodesic convolutional neural networks on riemannian manifolds." Proceedings of the IEEE international conference on computer vision workshops. 2015.*
>
> > *Boscaini, Davide, et al. "Learning shape correspondence with anisotropic convolutional neural networks." Advances in neural information processing systems 29 (2016).*
>
> > *Monti, Federico, et al. "Geometric deep learning on graphs and manifolds using mixture model cnns." Proceedings of the IEEE conference on computer vision and pattern recognition. 2017.*
>
> > *Hanocka, Rana, et al. "Meshcnn: a network with an edge." ACM Transactions on Graphics (ToG) 38.4 (2019): 1-12.*
>
> > *Bronstein, Michael M., et al. "Geometric deep learning: Grids, groups, graphs, geodesics, and gauges." arXiv preprint arXiv:2104.13478 (2021).*

---

### Meta-Review · Area_Chair_GPFr · 2023-12-12

**Metareview:**

The paper explores connections between various notions of convolutions over manifolds which have been introduced in the literature. It considers three coordinate based notions of geodesic convolution. The first, due to Masci et al, is a convolution operation expressed in geodesic polar coordinates, together with a maximum pooling operation to handle the ambiguity in the angular coordinate. The second, from a paper of Boscani et al, resolves this angular ambiguity through the anisotropic Laplacian, which is oriented with respect to the direction of maximum curvature at a given point. The third, due to Monti et. al. [M], generalizes the first two, through a parametric patch operator that uses a superposition of local “pseudo-coordinates”. The paper relates these three constructions to a differential geometric construction from the monograph of Bronstein et. al. [B], in which the coordinate ambiguity on tangent spaces is resolved through the choice of a gauge.

The paper’s main theoretical contribution relates these previous constructions to [B], through the introduction of a function that the paper calls a prior. This prior encodes the local geometry of the surface, and is learnable from data. The paper explores the space of valid priors, and presents experiments which show that the Dirac prior (which links previous constructions to that of [B]) leads to improved performance on benchmark datasets.

Reviewers raised a number of issues with the submission. Questions included algorithmic novelty (since the Dirac prior is best performing in experiments, and is the prior that links the constructions of [M] and [B]), the tightness of the connection between [M] and [B] (since the rephrasing involves a potentially continuous prior, rather than finitely many patches), clarity questions around the notion of the prior, and questions regarding the extensiveness of the experiments. While the work has explanatory value in clarifying the relationship between previously proposed surface convolution operations, it would benefit from revision to more clearly convey the role of the prior and the algorithmic implications of this connection.

**Justification For Why Not Higher Score:**

The work has explanatory value in clarifying the relationship between previously proposed surface convolution operations. At the same time, reviewers uniformly reported confusion regarding the paper's main contributions (the notion of a prior, the role of the Dirac prior, and the experiments), evaluating the work, in current form, as below the threshold.

**Justification For Why Not Lower Score:**

N/A

---

### Decision · Program_Chairs · 2024-01-16

Reject